# Decomposition of a set of distributions in extended exponential family form for distinguishing multiple oligo-dimensional marker expression profiles of single-cell populations and visualizing their dynamics

**Daigo Okada, Ryo Yamada**[ID]*

Unit of Statistical Genetics, Center for Genomic Medicine Graduate School of Medicine, Kyoto University, Kyoto, Japan

* ryamada@genome.med.kyoto-u.ac.jp

## Abstract

Single-cell expression analysis is an effective tool for studying the dynamics of cell population profiles. However, the majority of statistical methods are applied to individual profiles and the methods for comparing multiple profiles simultaneously are limited. In this study, we propose a nonparametric statistical method, called Decomposition into Extended Exponential Family (DEEF), that embeds a set of single-cell expression profiles of several markers into a low-dimensional space and identifies the principal distributions that describe their heterogeneity. We demonstrate that DEEF can appropriately decompose and embed sets of theoretical probability distributions. We then apply DEEF to a cytometry dataset to examine the effects of epidermal growth factor stimulation on an adult human mammary gland. It is shown that DEEF can describe the complex dynamics of cell population profiles using two parameters and visualize them as a trajectory. The two parameters identified the principal patterns of the cell population profile without prior biological assumptions. As a further application, we perform a dimensionality reduction and a time series reconstruction. DEEF can reconstruct the distributions based on the top coordinates, which enables the creation of an artificial dataset based on an actual single-cell expression dataset. Using the coordinate system assigned by DEEF, it is possible to analyze the relationship between the attributes of the distribution sample and the features or shape of the distribution using conventional data mining methods.

## Introduction

Single-cell expression analysis is an effective tool for studying the dynamics of cell population profiles [1–3]. Cytometry data, a type of single-cell expression data, quantify the amount of protein marker expression in each of a large number of randomly selected cells. Single-cell RNA sequencing (scRNA-seq) data, another type of single-cell expression data, has recently

**Data Availability Statement:** Data can be accessed in FlowRepository database (ID: FRFCM-ZYBC, FR-FCM-ZZZV, FR-FCM-ZZZU, and FR-FCM-ZZYA)

**Funding:** RY, grant numbers JPMJCR1502 and JPMJCR15G1. Core Research for Evolutional Science and Technology (CREST) URL of each funder website: https://www.jst.go.jp/kisoken/crest/en/. DO, grant number JP19J14816. KAKENHI Grant-in-Aid URL of each funder website: https://www.jsps.go.jp/english/e-grants/.

**Competing interests:** NO authors have competing interests.

become popular. This type of data allows comprehensive quantification of the amount of mRNA expression for genome-wide genes in single cells. Such single-cell expression data can be used to quantify or identify specific cell subsets based on the biomarkers. For example, specific lymphocyte subset (e.g. T cell and B cell subset) have been defined by the expression patterns of several cell surface protein markers [4, 5]. When many cells are sampled from a donor and their expression profiles are obtained, the expression data can be regarded as an observation of an unknown probability distribution of the cells. The expression profile of each cell can be viewed as a sample from a multidimensional distribution, where the number of dimensions is the number of markers.

Several computational methods developed for single-cell data analysis, such as spanning-tree progression analysis for density-normalized events (SPADE), monocle, and Wanderlust, have been used to investigate various phenomena [6–9]. Most of these methods focus on the diversity of multiple cells or the mutual phylogenic relationship among them within a set of cells sampled to identify subtypes of cells or to visualize their heterogeneity. Expression profile analyses such as cytometry and scRNA-seq are applied to many samples, each of which consists of many cells from individual donors. In recent years, demand for a computational method for heterogeneous multiple samples in the form of distribution has been increasing, and actually a method that integrates multiple expression profiles together and to identify subpopulation in data-driven manner was proposed [10]. These expression profiles take the form of multidimensional distributions, which have to be statistically investigated using methods such as clustering, case-control comparison, and chronological pattern analysis. Multiomics studies analyze phenotypes, transcriptomes, and cytometry data from hundreds or thousands of individuals [11–13]. In these studies, the distributions of cytometry profiles should be statistically analyzed with other datasets from different platforms. However, conventional statistical methods do not take distributions as inputs and thus cell population profiles in the form of distributions have to be modified into a suitable form, such as cell subtype fractions, via gating procedures. This modification of flow cytometry distribution data into multi-categorical fractions loses information. Therefore, the method used to convert the density information of a cell population into a form that can be handled by regular statistical procedures is very important.

Computational methods for extracting feature statistics from data in multidimensional distribution form can be classified into two types, namely parametric and nonparametric. A representative parametric method is the Gaussian mixture model [14]. This method is mainly used for the automation of the manual gating of cytometric data, which is of interest in computational cytometry. However, it is know that in many cases, the Gaussian mixture model, along with other parametric approaches such as t-mixture models [14], is too simple to represent the complexity of the distributions of a cell population profile.

Some nonparametric methods for embedding single-cell expression data or other kinds of distribution-type data into a low-dimensional space have been proposed [15–17]. Most of these methods are based on multidimensional scaling (MDS) [18]. MDS-based methods first estimate a population distribution based on samples using a nonparametric probability density estimation method such as the kernel density estimation method or the k-nearest neighbor (kNN) method [19, 20]. Then, the symmetric distance is defined between two distributions based on information theory. Finally, MDS-based methods generate a distance matrix for a set of distributions and embed the individual distributions in a low-dimensional Euclidean coordinate space that maintains these distance relationships as much as possible. Although this approach is simple and powerful for the visualization of samples from different donors, the embedding into Euclidean coordinate space is essentially non-precise and imperfect because the definition of distance based on information theory is non-Euclidean [21].

Information geometry is a field of statistics that deals with the geometry of probability distributions [21]. In this research, based on the idea of information geometry, we propose a method, called Decomposition into Extended Exponential Family (DEEF), for embedding sample distributions into a low-dimensional coordinate space. The DEEF method finds an exponential-family-like formula for an arbitrary set of distributions and component distributions to describe the set of distributions and gives the coordinates and potential function value for each distribution. The only difference between an extended exponential family (EEF) and the exponential family itself is that the potential function of a regular exponential family is convex whereas that of an EEF is not. DEEF estimates the inner products of distribution pairs and assigns coordinates $\theta$ to each distribution based on the eigenvalue decomposition of a matrix related to the inner products. The coordinate system contains imaginary coordinates, as in Minkowski space [22]. The coordinates $\theta$ can always recover the distributions without loss of information and in many cases, $\theta$ from only a limited number of principal axes can recover the original distributions with negligible residuals. This overcomes the drawbacks of the conventional MDS-based method.

In this paper, we define an EEF and discuss the theoretical aspects of the log-linear decomposition of the probability matrix $\mathbf{P}$ into exponential-like representations. We apply the DEEF method to a set of theoretical probability distributions and show that it can be used for data-driven extraction and the visualization of potential parameter structures of the dataset. We then apply DEEF to a cytometry dataset to examine the effects of epidermal growth factor (EGF) stimulation on an adult human mammary gland. It is shown that DEEF can extract parameters that identify the principal patterns of the cell population profile and describe the complex dynamics of cell population profiles as a trajectory. In addition, DEEF can be used to perform a dimensionality reduction for this dataset and a time-series reconstruction, which enables the creation of an artificial cytometry dataset based on the properties of the actual data.

## Results

### Method outline

We propose a statistical method called DEEF (Fig 1). An exponential family is a set of probability distributions whose probability density/mass functions are expressed in the form

$$\log P(x, \theta) = C(x) + \sum_{k=1} F_k(x)\theta_k - \psi(\theta) \tag{1}$$

where $C(x)$, $F_k(x)$, and $\psi(\theta)$ are known functions ($\psi(\theta)$ should be convex), and $\theta$ is the

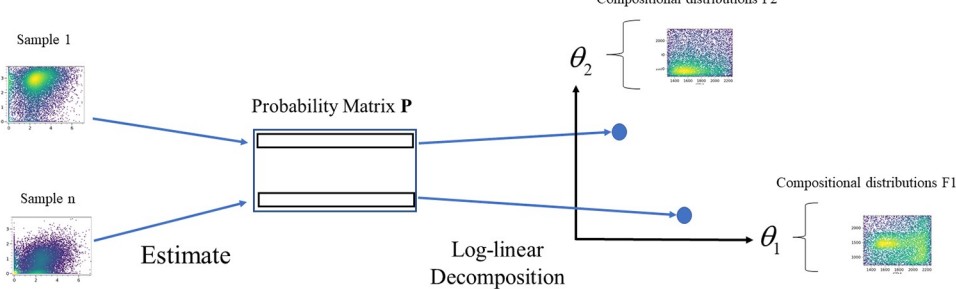

**Fig 1. Graphical outline of proposed method.** The outline of DEEF for embedding data from multiple distributions in the $\theta$ coordinate space with its compositional distribution $F$.

parameters that specify distribution instances. Many parametric probability distributions, such as the normal distribution and the binomial distribution, are included in the exponential family. Some probability distributions are not included in the exponential family, such as the mixture normal distribution. The details are given in S1 Text.

The distributions, one dimensional or multidimensional, in life sciences and other field, including expression profiles, are sometimes too complex to fit to simple parametric distribution. Some of them can be adequately described as a mixture of multiple parametric distributions. Actually, mixture of multiple distributions such as a mixture normal distribution or a mixture t distribution is commonly used in the parametric model for cytometry data [11]. And further complicated distributions can be fitted to only non-parametric distribution. Choosing the appropriate parametric model is difficult because it depends on the situation. While the exponential family can represent many simple probability distributions, it cannot represent most mixture distributions or more complex distributions often used in single-cell expression analysis.

We define an EEF as:

$$\log P(x, \theta) = C(x) + \sum_{k=1} F_k(x)\theta_k - \psi'(\theta) \tag{2}$$

$$\psi'(\theta) = \sum_{k=1} h_k \theta_k^2 \quad where \quad h_k = -1 \quad or \quad 1 \tag{3}$$

An EEF is almost identical to Eq 1, but with the potential function $\psi(\theta)$ modified as shown in Eq 3. We loosened the restriction that $\psi(\theta)$ should be convex so that a set of arbitrary distributions can fit the formula. We also modified $\psi(\theta)$ as shown in Eq 3. $\theta$ represents the coordinates of each distribution, where the inner product of the $\theta$ coordinates between two distributions is defined as half the logarithm of the inner product of density/mass functions. Using this definition of $\theta$, $C(x)$ and $F_k(x)$ are solvable when a set of distributions P(x, $\theta$) is given.

We obtain a set of multidimensional probability distributions from the experimental results. We divide the space into grid cells and estimate the probability mass functions P for the grid cells, which makes the dimensions of Eqs 2 and 3 finite and makes the estimation of EEF forms a linear algebraic calculation.

A matrix-operation-based simple algorithm can be constructed for log-linear decomposing probability matrix **P** into **C** + **ΘF** − **Ψ**, where **C**, **ΘF**, and **Ψ** are the discretized representations of EEF forms for multiple distributions (details given in Method section). Then, we can obtain the EEF representation of any distribution set. The input is only the probability matrix **P**, whose rows represent the probability mass function. DEEF can be applied to distribution sets to embed each distribution in the defined EEF space by considering **Θ** as the feature statistics of the distributions. Because the θ coordinate is calculated from eigenvalue decomposition, a few coordinates with the top eigenvalues contain a lot of the information of the probability distribution set. In addition, the **F** matrix provides principal compositional distributions in the original space. DEEF extracts the compositional distribution $F_i$ to a data driven manner. The $\theta_i$ coordinate indicates how much each sample has $F_i$. This is an interpretation of θ coordinate space, where hold difference between samples. A detailed description of the theory are given in the Appendix in S1 Text.

**Simulation data analysis.** First, we applied DEEF to a normal distribution set that consisted of 900 instances of a normal distribution, with the mean ranging from −1 to 1 and the standard deviation (sd) ranging from 2 to 4 at a fixed interval of 0.069 for each (Fig 2(a)). We called these parameters defined in the specific parametric models as original parameters. And,

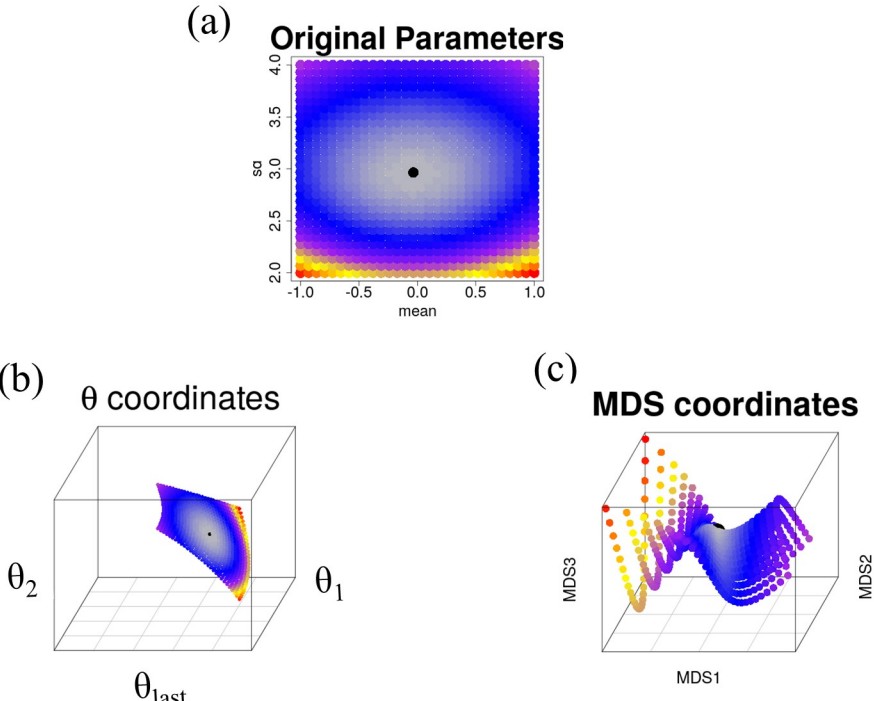

**Fig 2. Comparison of (a) original parameter space, (b) $\theta$ coordinate space, and (c) MDS coordinate space in normal set with the two parameters.** The theoretical KL-divergence-based distance from one member distribution (black point) is visualized by the color scale. The Euclidean distance in the original parameter space does not match the KL-divergence-based distance. The Euclidean distance in the MDS space approximates the KL-divergence-based distance, but the parameter structure is broken, unlike the case when embedding in the coordinate space.

a space using these original parameters as coordinate axes is called an original parameter space. We compared the DEEF method and a conventional MDS-based method [15] using this normal distribution set.

We compared the $\theta$ coordinate spaces with the top three absolute eigenvalues ($\theta_{last}$, $\theta_1$, $\theta_2$) (Fig 2(b)) and the top three MDS coordinate spaces (MDS1, MDS2, MDS3) (Fig 2(c)). The $\theta$ coordinate is denoted $\theta_i$ in decreasing order of eigenvalues. $\theta_{last}$ is the coordinate corresponding to the lowest eigenvalue, whose absolute value is largest in this case. Although both methods displayed a two-dimensional manifold in three-dimensional space, the two-dimensional manifold for DEEF was much simpler than that for MDS. The colors in Fig 2 indicate the Kullback-Leibler (KL) divergence from the distribution in the center of the mean-sd parametric grid (indicated by a black dot). Because the two-dimensional manifolds of DEEF and MDS were curved surfaces, it was not appropriate to use the Euclidean distance between points as a measure of divergence between two distributions. However, the simpler manifold for DEEF seems to be intuitively better for visualizing divergence. The number of total extracted coordinates for the MDS-based method was 445 because the decomposed matrix was not positive definite and some information was missing; the number of total extracted coordinates for DEEF was 900.

The normal distribution can usually be characterized by two parameters, mean and sd, on the original parameter space. However, they are also allowed to be expressed in different two parameters. While parameterization by mean and sd is only possible under the assumption that it is a normal distribution, the $\theta$ coordinates calculated by DEEF can be assigned to the

distribution without any assumptions. In both original parameters and $\theta$ coordinates, information about the difference between distributions is represented by the same number of parameters. In fact, when the distributions are generated sufficiently densely, it is visualized in Fig 2 that the topological relation among the distributions is maintained.

We apply DEEF to multiple normal distribution sets with different parameter structures, namely a mixture normal distribution set and an exponential distribution set, in S1 Text. Here, we apply the DEEF method to a set of theoretical probability distributions and show that it can be used for data-driven extraction and the visualization of the potential parameter structures of the dataset. DEEF successfully embedded these distributions in the $\theta$ coordinate space. The distributions could be recovered without loss of information and in many cases $\theta$ from only a limited number of principal axes could recover the original distributions with negligible residuals.

## EGF stimulation cytometry data analysis

Cytometry data can be considered as an unknown multidimensional probability distribution of cells, where the number of dimensions is the number of markers. We applied DEEF to a cytometry dataset.

We used mass cytometry data from a study on the effect of EGF stimulation on an adult human mammary gland [23]. In the experiment, measurements were made at 10 time points (0, 0.5, 1, 3, 6, 10, 30, 15, 60, and 120 minutes) in two replicates, one each after EGF stimulation and under control conditions. We picked four marker proteins, namely pAKT, pERK, pPLC$\gamma$2, and pS6, which were shown to respond to EGF stimulation in the original study. The pre-processed marker expression data for each time point after EGF stimulation for Replicate1 and Replicate2 are shown in Fig 3. We applied the DEEF method to the four marker single-cell expression datasets. Unlike for the simulation data, the population distribution was unknown and thus a sample set was obtained. Then, we estimated the probability matrix **P** of the single-cell expression dataset before we applied DEEF, as described below. Each single-cell expression dataset was a sample set from an unknown population distribution in the number-of-markers-dimensional space (four-dimensional space in this case). First, we decided the range of each marker. For each sample, we calculated the $\alpha$ percentile and the $1—\alpha$ percentile for each marker expression. We used the range of each marker between the minimum $\alpha$ percentile value and the maximum $1—\alpha$ percentile value among all samples so that all samples contained the expression range between the $\alpha$ and $1—\alpha$ percentiles for cells. In this case, we used $\alpha =$ 0.05. Next, we separated this range into equally spaced m points (m = 20), where m is a defined parameter. The number of grids was $m^4$. For the determined grids, we estimated the probability density using the kNN method (k = 800). The row vector P, representing the kNN-based densities of $m^4$ grids, was standardized so that its total value was 1. We applied the DEEF method to **P** built using the above procedure and calculated the corresponding $\theta$ coordinates. $\theta_{last}$ corresponded to a negative eigenvalue, and $\theta_1$, $\theta_2$, and $\theta_3$ corresponded to positive eigenvalues (S1 Fig). The boxplot of error shows that the performance of the distribution reproduction increases with increasing number of $\theta$ coordinates but at a slower rate than that for the simulation distribution set (S1 Fig).

We embedded all cell population profiles into a low-dimensional coordinate space and visualized them using the DEEF method. $\theta_1$, $\theta_2$, and $\theta_3$ accounted for 69.6%, 13.9%, and 8.9% of the sum of positive eigenvalues, respectively. Fig 4(a) shows scatter plots of the top positive $\theta$ coordinates derived from the DEEF method. $\theta_1$ and $\theta_2$ give common trajectories during EGF stimulation between Replicate1 and Replicate2 but $\theta_3$ gives a different trajectory. After EGF stimulation, the cell population profile moved on the $\theta_1$ and $\theta_2$ coordinate space and then

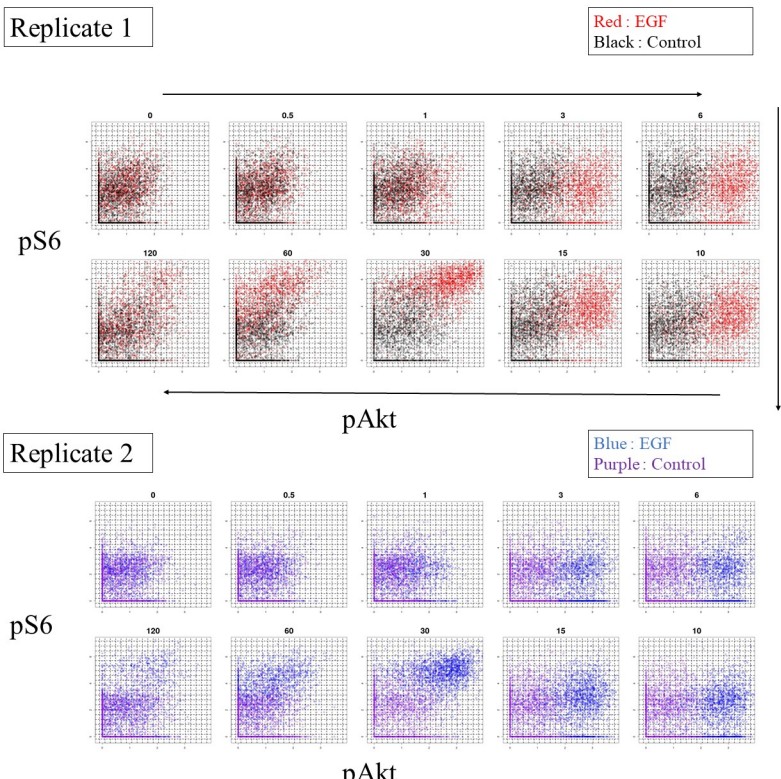

**Fig 3. Scatter plot of pAKT and pS6 at 10 time points after EGF stimulation.** For each replicate and condition, 2,000 randomly selected cells are plotted. The black dotted line represents the grids. The cell population profile changes dynamically after EGF stimulation but it is difficult to capture and evaluate this quantitatively using the raw data.

returned to the region near the baseline. We then used $\theta_1$ and $\theta_2$ to parameterize the cell population dynamics after EGF stimulation which is common between Replicates1 and Replicate2.

$F_1$ and $F_2$, which correspond to $\theta_1$ and $\theta_2$, respectively, show the type of cell population profile change represented by the trajectory. Fig 4(b) shows $F_1$ and $F_2$ for pAkt and pS6. $F_1$ explains the number of cells with high pAKT expression and high pS6 expression and $F_2$ explains the number of cells with low pAkt and high pS6 expression. An increase in $\theta_1$ and a decrease in $\theta_2$ correspond to the initial response. This change can be well expressed as a synthesis of the patterns of the three underlying cell population profiles. The increase in $\theta_2$ that occurs in the second half corresponds to the increase in pS6, which arose later than that of pAkt. The density plots of $F_1$ and $F_2$ for all four markers are shown in S2 Fig.

S3 Fig shows a scatter plot of all samples for MDS1 and MDS2 derived by applying the MDS-based method to this dataset. The dynamics after EGF stimulation have a trajectory pattern similar to that obtained with DEEF. However, we cannot get further information from this analysis.

To visualize F(x) as a four-dimensional function all at once, we performed SPADE analysis and described $F_1$ and $F_2$ on the SPADE tree. SPADE is a computational cytometry method that automatically clusters cells for multiple cytometry datasets and creates one consensus tree of the cell clusters. We applied SPADE to all 40 samples to create a SPADE tree that consisted of ten cell clusters (Fig 5(a)). Each SPADE cluster can be characterized by the four-marker expression pattern (Fig 5(b)). Fig 5(c) shows SPADE trees with $F_1$ and $F_2$ values. Each cluster was assigned $F_1$ and $F_2$ values of the grid to which the representative location of the cluster

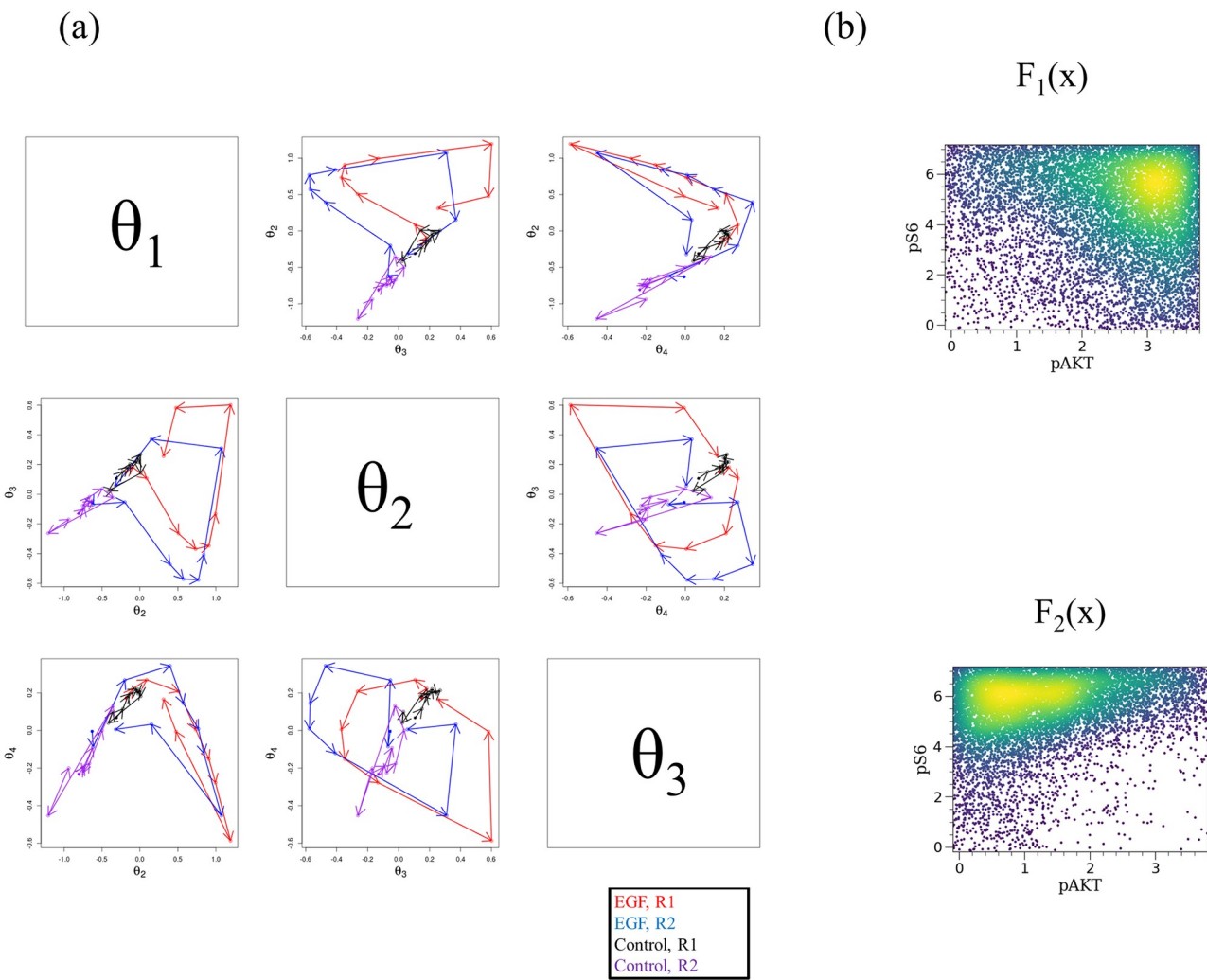

**Fig 4. Application of DEEF to EGF stimulation data.** The dynamics of the whole cell population profile are visualized and the dominant patterns that explain differences are extracted. (a) $\theta$ coordinate plot for coordinates $\theta_1$, $\theta_2$ and $\theta_3$ (i.e., those with the top positive eigenvalues). (b) $F_1$ and $F_2$ in DEEF for pAkt and pS6. The density plot was generated from 10,000 randomly sampled data points from the standardized $exp(F_i)$.

belongs. In $F_1$ on the SPADE tree, Cluster 9 has the highest positive $F_1$ values. This result is reasonable because Cluster 9 showed high expression for all four markers. This result corresponds to the fact that all marker expressions increase after EGF stimulation. Cluster 8 has the highest negative $F_1$ value, which is reasonable because this cluster showed low expression for all four markers. $F_2$, which corresponds to a different trajectory pattern from that for $F_1$, shows a different pattern on the SPADE tree. Cluster 3, which has the highest positive $F_2$ values, showed high expression for pS6 and pPLC$\gamma$2. These two markers are expressed later than pAkt and pERK. Interestingly, Cluster 2, which showed low expression for pERK and pS6, has the highest negative $F_2$ value. Using the table of the representative values for each cluster (S1 Table), this subset can be confirmed on the density plot of samples obtained 6 minutes after stimulation (Fig 5(d)). The DEEF method can provide insight into patterns that are difficult to detect using conventional methods.

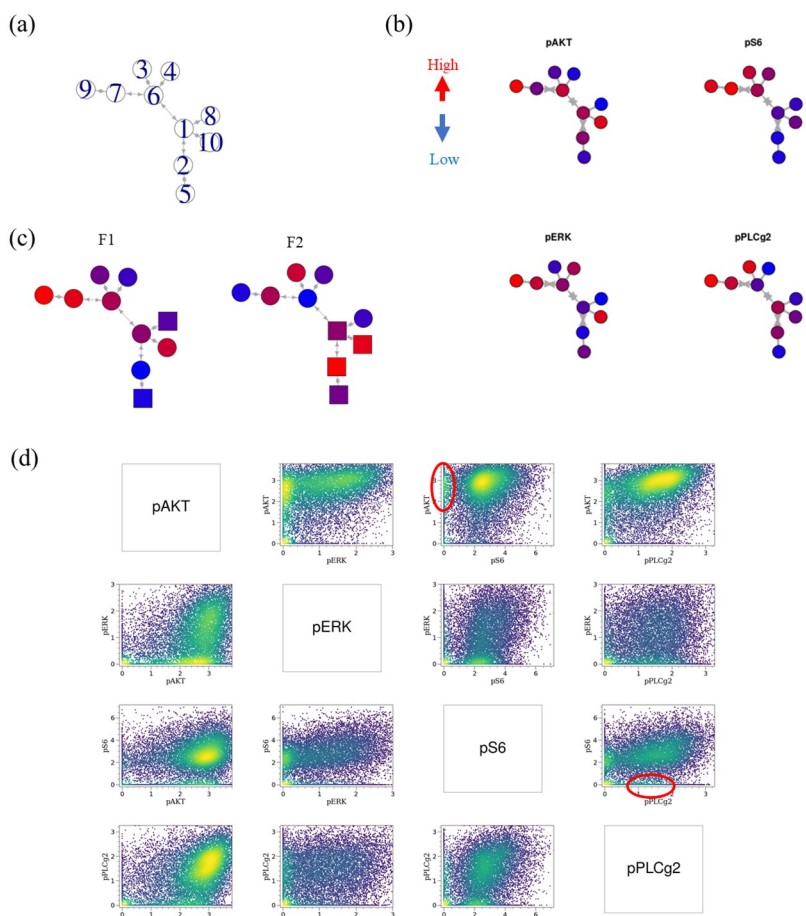

**Fig 5. $F_2$ and $F_3$ of EGF stimulation data on SPADE tree.** (a) Created SPADE tree with cluster number labels. (b) SPADE trees with four-marker expression. The color represents each marker expression value. (c) SPADE trees with $F_1$ and $F_2$ values. Each cluster was assigned $F_1$ and $F_2$ values of the grid to which the representative location of the cluster belongs. (d) Region of Cluster 2 of SPADE tree of EGF stimulation data. The corresponding regions of SPADE Cluster 2 are shown by a red circles in the density plots of the four markers obtained 6 minutes after EGF stimulation for Replicate1.

## Dimension reduction and time-course reconstruction using EGF stimulation dataset

In the previous section, we showed that DEEF works well with a real cytometry dataset. In this section, as further applications of DEEF for biological research, we describe dimensionality reduction and time-course reconstruction.

DEEF can reconstruct a distribution using only the coordinates with the top absolute eigenvalues. To reduce the dimensionality of a cell population profile, we expressed the cell population profile using only the synthetic sum of the main patterns; other differences were considered to be noise. A dimension reduction of the EGF stimulation dataset using the top $\theta$ coordinates was conducted. The panels in the first row of Fig 6(a) shows the change in the median marker intensity in the raw data along the time course for the four markers. The expression levels of pAKT and pERK increased first, followed by those of pS6 and pPLC$\gamma$2. This is consistent with the results in the original study. The panels in the second row of Fig 6 show the change in the median of marker intensity calculated from the reconstructed distribution using $\theta_1$, $\theta_2$, and $\theta_{last}$, corresponding to top three highest absolute eigenvalues (K = 3).

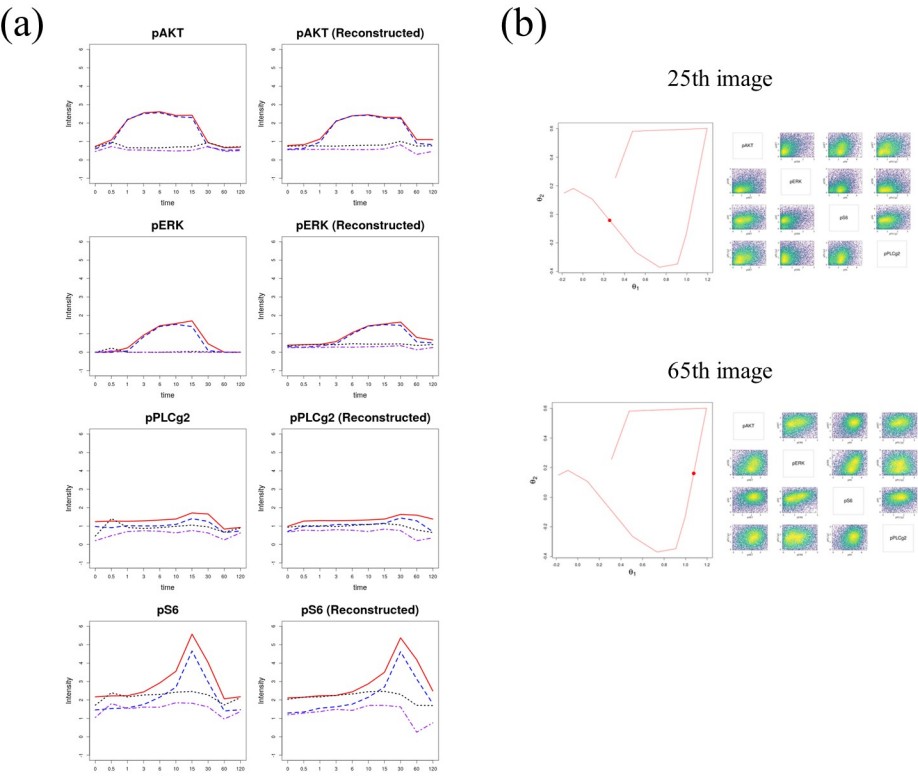

**Fig 6. Results of dimension reduction of cell population profiles using the DEEF method.** The reduction preserves the change in the marker expression along the time course for each marker (pAkt, pERK, pPLCγ2, and pS6). Left panels are the median values for each marker expression, which match those in the original study. Right panels are the median values for the distribution reproduced using the top three $\theta$ coordinates, namely $\theta_{last}$ with the highest negative eigenvalue and $\theta_1$ and $\theta_2$ with the highest positive eigenvalues (K = 3). (b) 25th and 65th images of 91 images as examples of the estimated cell population profiles between the measurements of Replicate1 after EGF stimulation. The corresponding points in the $\theta$ coordinate space are indicated by red dots.

These results suggest that the cell population profile reconstructed using only the main patterns well captures the characteristics of the dynamics of the original data. Here, the patterns that have a small contribution to the difference among the sample set were eliminated. If DEEF can decompose the information into meaningful data and noise, reproduction using only principal functions would denoise the data.

Next, using this scheme, we conducted a time-course reconstruction of Replicate1's EGF stimulation dataset whose original time course contained 10 time points. The value of the $\theta$ coordinate at each time point was estimated by linearly interpolating and dividing the value of the $\theta$ coordinate between each time point into 10 equal parts, and reconstructing the $\theta$ coordinate at a total of 91 images. Fig 6(b) shows the 25th and 65th images of the 91 images as examples of the estimated cell population profiles between the measurements. Based on the estimated value, the distribution was reproduced at K = 3. An animation of the cell population dynamics including the unmeasured time points is available (S1 Movie).

## Discussion

In this study, we proposed a class of probability distributions called EEFs and a nonparametric decomposition method for probability distribution sets called DEEF (Fig 1). The DEEF method provides geometric coordinates for each distribution and obtains feature statistics for

a sample set by estimating an exponential family-like representation for a multidimensional probability distribution set. DEEF can identify the parameters that well discriminate the difference among a distribution set as $\theta$. In addition, the coordinates identified by DEEF have a biological meaning, as shown by $F_i(x)$. The log-linear decomposition did not lose the information in the original datasets and the original distributions could be reproduced. The DEEF method extracted the feature statistics of distributions as $\theta$ coordinates without loss of information, unlike similar methods such as the MDS-based method (Fig 2, S1 Text).

When the DEEF method was applied to a cytometry dataset obtained after EGF stimulation, as shown in Fig 3, it extracted the main underlying patterns from the probability distribution set, embedded them into the coordinate system, and indicated the quantitative differences among samples (Fig 4). We parameterized the dynamics after the EGF stimulation with two parameters and expressed them as trajectories. We could then visualize the $F(x)$ function on the SPADE tree (Fig 5). By using SPADE, information on the combination of multidimensional markers can be simultaneously visualized; this is not possible with a two-marker density plot. The characteristics of the response to EGF are useful for characterizing a subset of human mammary cells and are essential information for understanding the properties of epithelial cancers [23]. DEEF may provide new insights into such characteristics with consideration of not only the change of a single marker but also a combination of multiple markers.

As a further application of DEEF, we performed a dimension reduction and a reconstruction of cell population profiles using highly contributing coordinates (Fig 6, S1 Movie). This method is considered to be effective for complementing cytometry data acquired along the time course. When cytometry data have an ordered structure such as a time series, complementary estimation of the state between measurements can be performed. In addition, DEEF can easily create an artificial dataset with a large sample size that conforms to the properties of the real data. This is useful in computational biology research.

In this study, cell population profiles were embedded into a low-dimensional space by applying the DEEF method to flow cytometry data. By treating the values of $\theta$ coordinates as a trait and performing an association analysis with genotype and transcriptome data, DEEF can identify genes and pathways related to the entire cell population profile and their dynamics. Multiomics analysis, which combines various types of large-scale omics data such as genomes, transcriptomes, and metabolomes, is widely used in various fields to study complex life systems [24–26]. Our research will make it easier to add single-cell data to multiomics analysis. In many biological fields, such as immunology and stem cell biology, the behavior of a whole cell population profile is very important for elucidating life phenomena. This behavior can be very complicated. A combination of the proposed method and omics analysis is expected to advance the understanding of these complex biological phenomena.

In recent years, high-dimensional single cell expression data such as scRNA-seq or CyTOF has become popular. Computational methods for such high-dimensional single cell expression data are also being actively developed [27]. On the other hand, DEEF is not suitable for handling genome-wide gene expression because the number of grids grows exponentially with the dimensionality and kNN estimation and the linear algebraic algorithm can't work well. However, by the novel theory and algorithm, DEEF provides high-resolution analysis for sample heterogeneity where the calculated coordinates and the original marker expression pattern are completely associated by F(x) function. In many case, cellular subsets, such as lymphocyte subset, have been defined by the expression patterns of several markers. From this perspective, DEEF are expected to provide a novel insight on the analysis of cell population profiles. Then, it is necessary to select only a few important markers for high-dimensional CyTOF and scRNA-seq data. Although choosing irrelevant markers would theoretically not have much effect on the results because DEEF treats each grid as independent, it would waste

computational resources. One potential solution might be the combination of DEEF with dimension reduction method, such as t-SNE and Uniform Manifold Approximation and Projection (UMAP) [28], although it seems necessary to study the effect of the non-linear embedding on the DEEF's decomposition logic. Further investigations would be beneficial to overcome this drawback.

Several other improvements can be considered for the DEEF method. In its present form, DEEF handles grids independently; it does not consider the positional relationships among neighboring grid cells. Taking these relationships into account would make the functions C and F smoother, which may remove random errors and improve machine learning accuracy and the interpretability of results. Another possible improvement is the use of the kernel method to estimate **P** from raw data. In the present procedure, DEEF calculates the inner products between distributions discretely using kNN density estimation. This step could be improved by embedding the dataset into a reproducing kernel Hilbert space with infinite dimensions directly using the kernel method [29]. The introduction of the kernel method into DEEF might improve performance.

## Conclusion

In this study, we developed a method called DEEF to analyze differences between cell population profiles using single-cell expression data. DEEF performs a log-linear decomposition of the probability matrix **P** to embed the distributions into a low-dimensional space. The DEEF method can extract the potential parameters of the probability distribution set and describe the meaning of the estimated parameters. Because single-cell expression data can be regarded as samples from an unknown population distribution, we can investigate the difference among cell population profile sets. DEEF can be used to examine and visualize the difference among single-cell expression datasets. DEEF can reconstruct the distributions from the top coordinates, which enables the creation of artificial datasets based on an actual single-cell expression dataset. Using the coordinate system assigned by DEEF, it is possible to analyze the relationship between the attributes of the distribution samples and the features or shape of the distribution using conventional data mining methods.

## Method

### 1. DEEF method

First, we describe the theoretical basis of DEEF. An exponential family is a set of probability distributions whose probability density/mass functions are expressed in the form:

$$\log P(x, \theta) = C(x) + \sum_{k=1} F_k(x)\theta_k - \psi(\theta) \tag{4}$$

where $C(x)$, $F_k(x)$, and $\psi(\theta)$ are known functions ($\psi(\theta)$ should be convex) and $\theta$ is the parameters that specify distribution instances. Many parametric probability distributions, such as the normal distribution and the binomial distribution, are included in the exponential family. Some probability distributions are not included in the exponential family, such as the mixture normal distribution. We define an EEF as:

$$\log P(x, \theta) = C(x) + \sum_{k=1} F_k(x)\theta_k - \psi'(\theta) \tag{5}$$

$$\psi'(\theta) = \sum_{k=1} h_k\theta_k^2 \quad where \quad h_k = -1 \quad or \quad 1 \tag{6}$$

where an EEF is almost identical to Eq 4, but with the potential function $\psi(\theta)$ modified as shown in Eq 5. We loosened the restriction that $\psi(\theta)$ should be convex so that a set of arbitrary distributions can fit the formula. We also modified $\psi(\theta)$ as shown in Eq 6. $\psi'(\theta)$ does not become a convex function unless $h_k$ is all 1. Therefore, an EEF can be defined as a probability distribution family that conditionally excludes rules on the convexity of the potential function from the definition of an exponential family.

Regardless of whether the potential function is convex or not, the functional inner product between exponentially expressed functions $P(x)$ and $Q(x)$ can be expressed as follows using only $\theta$ coordinates and the potential function (proof is shown in S1 Text, Appendix Theorem 1).

$$< P(x, \boldsymbol{\theta}^P), Q(x, \boldsymbol{\theta}^Q) >= \frac{e^{\psi(, \boldsymbol{\theta}^P + , \boldsymbol{\theta}^Q)}}{e^{\psi(, \boldsymbol{\theta}^P)} e^{\psi(, \boldsymbol{\theta}^Q)}} \tag{7}$$

If $P(x)$ and $Q(x)$ are both EEFs, the following simple relationship between $P(x)$ and $Q(x)$ is satisfied for their functional inner product and $\theta$ coordinates (proof is shown in S1 Text, Appendix Theorem 2).

$$\frac{1}{2} \log < P(x, \boldsymbol{\theta}^P), Q(x, \boldsymbol{\theta}^Q) >= \sum_{k=1} h_k \theta_k^P \theta_k^Q \tag{8}$$

Consider an $n \times n$ matrix $\mathbf{M}$, whose (i, j)-th element $m_{i,j}$ is identified as $\frac{1}{2} \log q_{i,j}$ where $q_{i,j}$ is the functional inner product between i-th and j-th distributions. Let the i-th eigenvalue of $\mathbf{M}$ be $\lambda_i$. Then, $\mathbf{M}$ can be represented by eigenvalue decomposition as follows:

$$\mathbf{M} = \mathbf{V}^T \boldsymbol{\Lambda} \mathbf{V} \tag{9}$$

where the i-th column of $\mathbf{V}$ represents the i-th eigenvectors of $\mathbf{M}$ and $\boldsymbol{\Lambda}$ is a diagonal matrix whose i-th diagonal elements are $\lambda_i$. Note that the eigenvalues of $\mathbf{M}$ contain negative values. Then, $\mathbf{M} = \mathbf{V}^T \boldsymbol{\Lambda}' \mathbf{S} \mathbf{V} = (\mathbf{V}\sqrt{\boldsymbol{\Lambda}'})^T \mathbf{S} (\mathbf{V}\sqrt{\boldsymbol{\Lambda}'})$, where $\mathbf{S}$, $\boldsymbol{\Lambda}'$ and $\sqrt{\boldsymbol{\Lambda}'}$ are $n \times n$ diagonal matrices whose i-th diagonal elements are $sign(\lambda_i)$, $|\lambda_i|$, and $\sqrt{|\lambda_i|}$, respectively. Therefore, when we take the $\theta$ coordinate matrix $\boldsymbol{\Theta}$ and $h_i$ as follows, Eq 4 is completely satisfied.

$$\boldsymbol{\Theta} = \mathbf{V}\sqrt{\boldsymbol{\Lambda}'} \tag{10}$$

$$h_i = sign(\lambda_i) \tag{11}$$

where $\boldsymbol{\Theta}$ is the $\theta$ coordinate matrix whose (i,j)-th element represents the j-th coordinate value of the i-th distribution in the EEF expression. Because $\mathbf{M} = \boldsymbol{\Theta}^T \mathbf{S} \boldsymbol{\Theta}$, Eq 4 is completely satisfied.

The next step is the calculation of C(x) and $F_i$(x). To treat this calculation discretely using a computer, the above expression must be expressed in matrix form as:

$$\mathbf{P}^{log} = \mathbf{C} + \boldsymbol{\Theta} \mathbf{F} - \boldsymbol{\Psi} \tag{12}$$

where $\mathbf{P}^{log}$ is an $n \times m$ matrix that represents a log-discretized probability mass function of m grids of n samples, $\mathbf{C}$ is an $n \times m$ matrix that corresponds to C(x) and all of whose rows have the vector $\mathbf{c}$, $\boldsymbol{\Theta}$ is the $n \times n$ matrix obtained previously, $\mathbf{F}$ is an $n \times m$ matrix whose row vector corresponds to discretized $F_i(x)$, and $\boldsymbol{\Psi}$ is an $n \times m$ matrix whose column vector is the previously obtained $\sum_{k=1} h_k \theta_k^2 \mathbf{1}$. Then, this equation is rewritten as:

$$\mathbf{P}' = \boldsymbol{\Theta}' \mathbf{F}' \tag{13}$$

where $\mathbf{P}' = \mathbf{P}^{\mathbf{log}} + \boldsymbol{\Psi}$, $\mathbf{F}'$ is $[\mathbf{F}^T, \mathbf{c}]^T$, and $\boldsymbol{\Theta}'$ is $[\boldsymbol{\Theta}, \mathbf{1}]$. Therefore, $\mathbf{F}'$ can be obtained using the

Moore-Penrose pseudo-inverse matrix $Ginv(\mathbf{\Theta}')$ as follows:

$$\mathbf{F}' = Ginv(\mathbf{\Theta}')\mathbf{P}' \qquad (14)$$

Because $\mathbf{F}'$ is defined as $[\mathbf{F}^T, \mathbf{c}]^T$, all items necessary for the EEF expression of the distribution set can be obtained.

Based on the above theory, it is possible to construct a simple matrix-operation-based algorithm for decomposing probability matrix $\mathbf{P}$ to obtain the EEF representation of any distribution set. The input is probability matrix $\mathbf{P}$, whose rows represent the probability mass function. The first step is calculating matrix $\mathbf{M}$ from $\mathbf{P}$. The second step is the eigenvalue decomposition of $\mathbf{M}$. $h_i$ are obtained to determine $\psi'(\theta)$ and an n sample × n coordinate matrix $\mathbf{\Theta}$ is obtained to embed all samples. The third step is calculating $\mathbf{c}$ and $\mathbf{F}$ to determine all components of the EEF expression. The simulation data analysis method is described in S1 Text.

This method can be applied to distribution sets to embed each distribution in the defined EEF space by considering $\mathbf{\Theta}$ as the feature statistics of the distributions. Because the $\theta$ coordinate is calculated from eigenvalue decomposition, a few coordinates with the top eigenvalues have a lot of the information of the probability distribution set. In addition, the $\mathbf{F}$ matrix provides principal compositional distributions in the original space. The R package "deef" is available on GitHub (https://github.com/DaigoOkada/deef).

## 2. Distribution reproduction and performance evaluation

In DEEF, the distribution can be reproduced using any number of coordinates when C, $F_i$, $\theta_i$, and $h_i$ are obtained. We reproduced the distribution by reconstructing the probability mass function calculated by normalizing $exp(C(x) + \sum_{i=1}^{K} F_i(x)\boldsymbol{\theta}_i - \psi(\boldsymbol{\theta}))$, where the coordinates with the top K absolute eigenvalues were selected. In this study, performance was evaluated by Performance Index (PI) defined by the sum of the squared error between the true probability mass function and the reconstructed probability mass function. This value was calculated for each distribution included in the distribution set. A smaller squared error indicates better reproduction. In particular, if this value is zero, the original distribution and the reconstructed distribution are exactly the same.

## 3. Conventional MDS-based method

We embedded the distribution set using an MDS-based method using the following procedure. First, we calculated the distance matrix among samples. The distance between two distributions $p_i$ and $p_j$ is defined as $\frac{1}{2}(KL(p_i||p_j) + KL(p_j||p_i))$, and the coordinate values of each sample are calculated by applying MDS to the generated distance matrix. MDS was applied to this distance matrix to calculate the MDS coordinates of each sample. The coordinates are denoted MDS1, MDS2 and MDS3 in descending order of their eigenvalues.

## 4. Application of DEEF method to normal distribution set

We applied DEEF to a normal distribution set that consisted of 900 instances of a normal distribution, with the mean ranging from −1 to 1 and sd ranging from 2 to 4 at a fixed interval of 0.069 for each. The $\theta$ coordinate values and MDS were calculated using the theoretical value of the functional inner product or KL divergence defined by the mean and sd.

As the notation to distinguish the original parameter and $\theta$ coordinates, we named the original parameters using the alphabetic name used in the original parametric model. For example, in the case of normal distribution set, the original parameter is named as "mean" and "sd". On

the other hand, $\theta$ coordinates are always named as $\theta_i$ using the Greek letter $\theta$ and the suffix number $i$.

## 5. Construction of probability matrix P from single-cell expression dataset

Unlike for the simulation data, the population distribution was unknown and thus a sample set was obtained. We estimated the probability matrix **P** of the single-cell expression dataset before we applied DEEF, as described below. Each single-cell expression dataset was a sample set from an unknown population distribution in d-dimensional space, where d is the number of markers of the samples. First, we decided the range of each marker. For each sample, we calculated the $\alpha$ percentile and 1—$\alpha$ percentile of each marker expression. We used the range of each marker between the minimum $\alpha$ percentile value and maximum 1—$\alpha$ percentile value among all samples so that all samples contained the expression range between the $\alpha$ and 1—$\alpha$ percentiles for cells. Next, we separated this range into equally spaced m points, where m is a defined parameter. The number of grids is $m^d$. For the determined grids, we estimated the probability density using the kNN method. The row vector P, representing the kNN-based densities of $m^d$ grids, was standardized so that the sum of the vector was 1.

## 6. Application of DEEF method to EGF stimulation data

We used mass cytometry data from research on the effect of EGF stimulation on an adult human mammary gland [23]. The data were obtained from the Flow Repository (ID: FR-FCM-ZYBC). In the experiments, measurements were made at 10 time points (0, 0.5, 1, 3, 6, 10, 30, 15, 60, and 120 minutes) in two replicates after EGF stimulation and control conditions, respectively. We picked four marker proteins, namely pAKT, pERK, pS6, and pPLC$\gamma$2, which were shown to respond to EGF stimulation in the original study. As preprocessing, the marker expression levels were converted using asinh (intensity/5), as done in the original study. The number of cells in this dataset was between 8,089 and 22,221. We constructed probability matrix **P** from the cytometry data. Each cell could be taken as a sample from the population distribution. The hyperparameters for constructing **P** were m = 20, $\alpha$ = 0.05, and k = 800. Next, the DEEF method was applied to estimate **P**. The coordinates are denoted $\theta_1$, $\theta_2 \cdots \theta_{last}$ in descending order of their eigenvalues.

## 7. Visualization of F function with density plot and SPADE

We expressed $F_i$ as a compositional distribution by standardizing $exp(F_i)$ so that its total value was 1. Then, from this distribution, we sampled 10,000 data points and drew the density plot using the matplotlib Python library.

To visualize the multimarker information simultaneously, we applied the SPADE algorithm to the EGF stimulation data [6]. The number of clusters was 10 and other hyperparameters were the same as those in the original article. In Creating minimum spanning tree step, we used the mst function of R package "ape". We used the complete linkage method in the clustering step. The representative marker expression was the median values of the cells belonging to each cluster on the consensus tree.

## 8. Dimension reduction and time-course reconstruction of EGF stimulation data

DEEF can reconstruct a distribution using only the top coordinates. To reduce the dimensionality of a cell population profile, we expressed the cell population profile using only the synthetic sum of the main patterns; other differences were considered to be noise. The

reconstructed distributions (K = 3) were obtained using the procedure described in Method section 2. For each of the four markers (pAKT, pERK, pS6, and pPLCγ2), we visualized the median expression value change for the original marker expression and the reconstructed marker expression. For the original marker expression, for each sample, we calculated the median value of each marker from the expression value of cells. For the reconstructed marker expression, we integrated the reconstructed distribution and eliminated all markers (three) except the one that we focused on. Then, the 50th percentile value of the one marker expression was estimated as the median by linearly interpolating the values between the grids.

Next, we conducted the time-course reconstruction of Replicate1's EGF stimulation dataset whose original time course contained 10 time points. The value of the $\theta$ coordinate at each time point was estimated by linearly interpolating and dividing the value of the $\theta$ coordinate between each time point into 10 equal parts, and reconstructing the $\theta$ coordinate at a total of 91 time points. Based on the estimated value, the distribution was reproduced at K = 3.

## Supporting information

**S1 Text. Theory of DEEF and simulation data analysis.**
(PDF)

**S1 Fig. Comparison of DEEF and MDS-based method with EGF stimulation data.** (a) $\theta$ coordinate plot for coordinates $\theta_1$ and $\theta_2$ and (b) MDS coordinate plot for two coordinates MDS1 and MDS2 with the top eigenvalues.
(TIF)

**S2 Fig. 4-by-4 density plot of $F_1$ and $F_2$ for EGF stimulation data.**
(TIF)

**S3 Fig. Performance of DEEF for EGF stimulation data.** (a) Eigenvalue plots for an EGF stimulation dataset. Left panel shows the absolute eigenvalues standardized so that its total value was 1, where black bars are positive eigenvalues and white bars are negative eigenvalues. Right panel shows the cumulative sum of absolute eigenvalues. (b) Performance boxplot of distributions reconstructed using only the top K coordinates with high absolute eigenvalues for the EGF stimulation dataset. The performance was evaluated by the Performance Index (PI) defined by the sum of the squared error between the true probability mass function and the reconstructed probability mass function. The overall performance increases with increasing value of K.
(TIF)

**S1 Table. Representative marker expression values of ten clusters on the SPADE tree.**
(CSV)

**S1 Movie. Animation of cell population dynamics for 91 time points after EGF stimulation for Replicate1.** The reconstruction was done with $\theta_1$, $\theta_2$, and $\theta_{last}$ (K = 3).
(GIF)

## Acknowledgments

We would like to thank Prof. Masaru Ishii, Dr. Takao Sudo, and Dr. Tetsuo Hasegawa, who are members of the Department of Immunology and Cell Biology, Osaka University Graduate School of Medicine.

## Author Contributions

**Conceptualization:** Daigo Okada, Ryo Yamada.

**Data curation:** Daigo Okada.

**Formal analysis:** Daigo Okada.

**Funding acquisition:** Daigo Okada, Ryo Yamada.

**Investigation:** Daigo Okada.

**Methodology:** Daigo Okada, Ryo Yamada.

**Project administration:** Ryo Yamada.

**Software:** Daigo Okada.

**Supervision:** Ryo Yamada.

**Validation:** Daigo Okada.

**Visualization:** Daigo Okada.

**Writing – original draft:** Daigo Okada.

**Writing – review & editing:** Ryo Yamada.

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
