## [Decision Letter · Decision Letter 0]

4 Feb 2020

PONE-D-20-01313

Decomposition of arbitrary sets of distributions in extended exponential family form for distinguishing multiple expression profiles of single-cell populations and visualizing their dynamics

PLOS ONE

Dear Prof. Yamada,

Thank you for submitting your manuscript to PLOS ONE. After careful consideration, we feel that it has merit but does not fully meet PLOS ONE’s publication criteria as it currently stands. Therefore, we invite you to submit a revised version of the manuscript that addresses the points raised during the review process.

We would appreciate receiving your revised manuscript by Mar 20 2020 11:59PM. To enhance the reproducibility of your results, we recommend that if applicable you deposit your laboratory protocols in protocols.io, where a protocol can be assigned its own identifier (DOI) such that it can be cited independently in the future. For instructions see: http://journals.plos.org/plosone/s/submission-guidelines#loc-laboratory-protocols

We look forward to receiving your revised manuscript.

Kind regards,

Alan D Hutson

Academic Editor

PLOS ONE

Journal Requirements:

Additional Editor Comments (if provided):

Please attend to the major concerns of both reviewers. If these concerns are not addressed this manuscript will not be processed further. As noted by both reviewers there is strong potential for your methods, but there needs to be some additional non-trivial work prior to publication.

Reviewers' comments:

Reviewer's Responses to Questions

**Comments to the Author**

1. Is the manuscript technically sound, and do the data support the conclusions?

Reviewer #1: Yes

Reviewer #2: Yes

2. Has the statistical analysis been performed appropriately and rigorously? 

Reviewer #1: Yes

Reviewer #2: I Don't Know

3. Have the authors made all data underlying the findings in their manuscript fully available?

Reviewer #1: Yes

Reviewer #2: Yes

4. Is the manuscript presented in an intelligible fashion and written in standard English?

Reviewer #1: Yes

Reviewer #2: Yes

5. Review Comments to the Author

Reviewer #1: In this work, the authors proposed an interesting method (DEEF) for the decomposition of a set of exponential family distributions. The method makes use of the property of the inner product of extended exponential families (EEFs) and translates the decomposition of distributions into that of EEF parameters, which can be carried out by a standard spectral decomposition approach. The theoretical properties have been well discussed. The authors proposed the application of DEEF in the study of the expression profiles of single-cell populations, and it was demonstrated using a cytometry data set with four selected markers. The manuscript is mostly well written. Implementation has been shared online. However, I have a major concern regarding the proposed method and some minor comments.

Major comment

(1) High-dimensional single cell measurements are becoming increasingly common, such as CyTOF and single cell RNA-seq. The authors have discussed application in multi-omics. However, the proposed method relies on estimating the probability mass on a grid of the sample space through kNN or kernel density estimation. This will be quite difficult for high-dimensional settings for several reasons: 1. the number of grids grows exponentially with the dimensionality; 2. local methods such as kNN do not work well in high dimensional settings; 3. It is hard to select the parameters such as k and bandwidth. Therefore, I think the proposed method will have limited application for more recent single cell expression data. I recommend the authors demonstrate its application in a dataset with larger number of variables.

Minor comments

(2) Please elaborate on the interpretation of theta space. While it has been explained that the theta space is obtained by decomposition of n distributions, the authors can comment more on the relationship between theta space and the original parameters, and distinguish them by notations.

(3) The authors have shared the implementation on GitHub with an example starting from the probability mass matrix. It would be more helpful if the authors can provide the original data and include the code for obtaining the probability mass matrix.

(4) Please comment more in the main text on application of the method when the expression profile of a sample is a mixture of multiple distributions, which would not be exponential family but commonly encountered in single cell expression data.

(5) In application to EGF stimulation dataset, four markers known responding to the simulation have been selected. Will presence of irrelevant markers affect the method’s performance?

(6) Page 5, Line 172 and supplementary text Figure A, F. Please indicate the measure of performance and add labels to the y-axis.

(7) Page 10, Line 344~345, please denote explicitly log q_i,j = log (inner product of P(x, theta^P), Q(x, theta^Q)).

(8) Supplement Page 3, Figure A. The order of figures and the legend description is not consistent. It should be “2D, Random, 1D, Mixture”

(9) Supplement Page 4, Line 5 in main text. Correct “… a larger a value of…” to “a larger value of …”

(10) Supplement Page 5, Figure C. What is the x-axis in the plots and why is its support [0,1]? Please also label the y-axis.

Reviewer #2: A surely useful algorithm with a few practical shortcomings:

1) How does this work complement or expand on other algorithms offering similar approaches but more complex (RNA) data sets, for example Barkas, Nature Methods Aug. 2019?

2) The authors discuss that their current workflow only works on relatively low dimensional data. Wetlab methods move away from low dimension towards more complex data, such as scRNAseq or high dimensional maps of tissues or mass cytometry of complex blood and tissue samples. In order for this work to be applicable to a broader public the authors must use more complex data and also combine their analysis with dimension reduction methods such as UMAP (Uniform Manifold Approximation and Projection). UMAP runs very fast and could address many performance issues mentioned. Also UMAP, controversially to SPADE or the herein used trees, keeps single cells while giving the data a direction in form of population development or development over time.

6. PLOS authors have the option to publish the peer review history of their article (what does this mean?). If published, this will include your full peer review and any attached files.

Reviewer #1: No

Reviewer #2: Yes: Carsten Krieg, PhD; Department of Immunology & Dermatology, Medical University of South Carolina, SC 29425

---

## [Author Response · Author response to Decision Letter 0]

25 Feb 2020

To Editor

Besides the points suggested by the reviewers, we modified the legend of S4 Fig and Fig B in S1 Text to clarify the scale of absolute eigenvalues of distributional decomposition .

#### Changes ######

(S1 Text, Page4, Legend of Fig B) The left panel shows the absolute eigenvalues standardized so that its total value was 1, where the black bars are positive eigenvalues and the white bars are negative eigenvalues.

(Page16, Legend of S4 Fig) Left panel shows the absolute eigenvalues standardized so that its total value was 1, where black bars are positive eigenvalues and white bars are negative eigenvalues. 

##################

Comments from Reviewer 1

Thank you for the useful suggestions. We have highlighted the changes within the manuscript. The changed parts in the manuscript are also shown below each response.

Major comment(1): High-dimensional single cell measurements are becoming increasingly common, such as CyTOF and single cell RNA-seq. The authors have discussed application in multi-omics. However, the proposed method relies on estimating the probability mass on a grid of the sample space through kNN or kernel density estimation. This will be quite difficult for high-dimensional settings for several reasons: 1. the number of grids grows exponentially with the dimensionality; 2. local methods such as kNN do not work well in high dimensional settings; 3. It is hard to select the parameters such as k and bandwidth. Therefore, I think the proposed method will have limited application for more recent single cell expression data. I recommend the authors demonstrate its application in a dataset with larger number of variables.

Response: Thank you very much for this valuable comment and detailed suggestions. We totally agree with you. Unfortunately, the principal novelty of our manuscript is to propose a new method to decompose a set of data samples in the form of distribution in data-driven fashion, that is applicable to relatively low dimensional single cell expression profiles, but not to high-dimensional transcriptome data sets. We believe the title and abstract were misleading and the readers would be misled and wrongly anticipate our method could be applicable to single cell transcriptome analysis, as the you did. We feel sorry for this misleading wordings. Therefore we changed the title to “Decomposition of a set of distributions in extended exponential family form for distinguishing multiple oligo-dimensional marker expression profiles of single-cell populations and visualizing their dynamics” and we revised the Abstract and Introduction to clarify this point. Also we added comments on the importance of development of the method that is applicable to high-dimensional cases instead of applying our method to those data sets in Discussion to clarify this point. 

The higher dimensional expression profiles such as CyTOF or scRNA-seq, are increasingly more important as suggested. We agree with you and we commented their importance and the limitation of our method in the direction in Discussion Section.

######### Changes #############

(Page1, Abstract)In this study, we propose a nonparametric statistical method, called Decomposition into Extended Exponential Family (DEEF), that embeds a set of single-cell expression profiles of several markers into a low-dimensional space and identifies the principal distributions that describe their heterogeneity.

(Page1, Introduction, line8-line11) Such single-cell expression data can be used to quantify or identify specific cell subsets based on the biomarkers. For example, specific lymphocyte subset (e.g. T cell and B cell subset) have been defined by the expression patterns of several cell surface protein markers [4,5].

(Page9, Discussion, line321-line340) In recent years, high-dimensional single cell expression data such as scRNA-seq or CyTOF has become popular. Computational methods for such high-dimensional single cell expression data are also being actively developed [27]. On the other hand, DEEF is not suitable for handling genome-wide gene expression because the number of grids grows exponentially with the dimensionality and kNN estimation and the linear algebraic algorithm can't work well. However, by the novel theory and algorithm, DEEF provides high-resolution analysis for sample heterogeneity where the calculated coordinates and the original marker expression pattern are completely associated by F(x) function.In many case, cellular subsets, such as lymphocyte subset, have been defined by the expression patterns of several markers. From this perspective, DEEF are expected to provide a novel insight on the analysis of cell population profiles. Then, it is necessary to select only a few important markers for high-dimensional CyTOF and scRNA-seq data. Although choosing irrelevant markers would theoretically not have much effect on the results because DEEF treats each grid as independent, it would waste computational resources. One potential solution might be the combination of DEEF with dimension reduction method, such as t-SNE and Uniform Manifold Approximation and Projection (UMAP) [28], although it seems necessary to study the effect of the non-linear embedding on the DEEF’s decomposition logic. Further investigations would be beneficial to overcome this drawback.

Several other improvements can be considered for the DEEF method. In its present form, DEEF handles grids independently; it does not consider the positional relationships among neighboring grid cells.

################## 

Minor comments (2): Please elaborate on the interpretation of theta space. While it has been explained that the theta space is obtained by decomposition of n distributions, the authors can comment more on the relationship between theta space and the original parameters, and distinguish them by notations.

Response: DEEF extracts the composition distribution F_i in a data driven manner. The θ_i coordinate indicates how much each sample has F_i component. This is an interpretation of θ coordinate space. For example, the normal distribution is parameterized with two parameters, mean and sd^2, on the original parameter space. It is also parameterized with natural parameters of exponential family form, mean/sd^2 and -1/(2 sd^2). The θ coordinates calculated by DEEF correspond to the natural parameters of exponential family but θ coordinates are determined in the frame of extended definition of exponential family form and also θ coordinates are calculated in data-driven way; in other words, θ coordinates of one sample should be different when it is evaluated with different data set. As described here, θ coordinates vary in the context. However the number of meaningful θ coordinate axes corresponds to the degree of freedoms of the samples or the dimension of the information manifold where the samples should be localized. In the cases shown in Fig 2, many distribution instances are visualized and what the figure indicates is that the arrangement of instances kept their topological relation, although the coordinate systems were transformed. We added this topic in the Simulation data analysis sub-section in Result section. The added parts in the text are the following parts highlighted in red.

######### Changes #############

(Page4, Result, line129-line131) DEEF extracts the composition distribution F_i to a data driven manner. The θ_i coordinate indicates how much each sample has F_i. This is an interpretation of θ coordinate space, where hold difference between samples.

(Page5, Result, line156-line164) The normal distribution can usually be characterized by two parameters, mean and sd, on the original parameter space. However, they are also allowed to be expressed in different two parameters. While parameterization by mean and sd is only possible under the assumption that it is a normal distribution, the θ coordinates calculated by DEEF can be assigned to the distribution without any assumptions. In both original parameters and θ coordinates, information about the difference between distributions is represented by the same number of parameters. In fact, when the distributions are generated sufficiently densely, it is visualized in Fig 2 that the topological relation among the distributions is maintained.

########################

Response: As the notation to distinguish the original parameter and θ coordinates, we named the original parameters using the alphabetic name used in the original parametric model. For example, in the case of normal distribution set, the original parameter is named as “mean” and “sd”. On the other hand, θ coordinates are always named as θi using the Greek letter θ and the suffix number i. Because the notation of original parameter space is insufficient, we added this Result section. In addition, we added the explanation to distinguish original parameters and θ coordinate to Method Section.

######### Changes #############

(Page4, Result, line136-line139) We called these parameters defined in the specific parametric models as original parameters. And, a space using these original parameters as coordinate axes is called an original parameter space.

(Page412 Method, line439-line443) As the notation to distinguish the original parameter and $\\theta$ coordinates, we named the original parameters using the alphabetic name used in the original parametric model. For example, in the case of normal distribution set, the original parameter is named as “mean” and “sd”. On the other hand, θ coordinates are always named as θ_i using the Greek letter θ and the suffix number i.

##############################

Minor comments (3): The authors have shared the implementation on GitHub with an example starting from the probability mass matrix. It would be more helpful if the authors can provide the original data and include the code for obtaining the probability mass matrix.

Response: Since it took a time to download EGF stimulation dataset used in this research and construct P from it, we added the sample code to create P using GvHD dataset in Github page, which is built-in dataset FlowCore package. We plan to continually update this package and document. (Github: https://github.com/DaigoOkada/deef)

Minor comments (4): Please comment more the main text on application of the method when the expression profile of a sample is a mixture of multiple distributions, which would not be exponential family but commonly encountered in single cell expression data.

Response: I mentioned the comment about this topic as below.

######### Changes #############

(Page3, Result, line97-line106) The distributions, one dimensional or multidimensional, in life sciences and other field, including expression profiles, are sometimes too complex to fit to simple parametric distribution. Some of them can be adequately described as a mixture of multiple parametric distributions. Actually, mixture of multiple distributions such as a mixture normal distribution or a mixture t distribution is commonly used in the parametric model for cytometry data [11]. And further complicated distributions can be fitted to only non-parametric distribution. Choosing the appropriate parametric model is difficult because it depends on the situation. While the exponential family can represent many simple probability distributions, it cannot represent most mixture distributions or more complex distributions often used in single-cell expression analysis.

####################################

Minor comments (5): In application to EGF stimulation dataset, four markers known responding to the simulation have been selected. Will presence of irrelevant markers affect the method’s performance?

Response: Choosing irrelevant markers would theoretically not have much effect on the results because DEEF treats each grid as independent. We preliminary evaluated the effect of non-informative additional marker by adding a gene with Gaussian random expression to the EGF data. The output of DEEF to this data set with additional noise marker did not affect the result of DEEF and the similar chronological trajectories were identified (data not shown). Although the presence of irrelevant markers would not have much effect on the results, selecting markers are important step when using DEEF because choosing irrelevant markers would waste computational resources. We added this topic in the same paragraph with the response to major comment(1).

### Changes #########

(Page9, Discussion, line333-line335)Although choosing irrelevant markers would theoretically not have much effect on the results because DEEF treats each grid as independent, it would waste computational resources.

##############

Minor comments (6):Page 5, Line 172 and supplementary text Figure A, F. Please indicate the measure of performance and add labels to the y-axis.

Response: We added the measure of performance and labels in the legend of S4 Fig, Supplementary text Figure A, F. In addition, we added the term of Performance Index (PI) in the method section.

### Changes #########

(Page16, Legend of S4 Fig) The performance was evaluated by the Performance Index (PI) defined by the sum of the squared error between the true probability mass function and the reconstructed probability mass function. 

(S1 Text, Page3, Legend of Fig A) The fourth column panels show boxplots of the Performance Index (PI) defined by the sum of the squared error of distributions reconstructed using only the top K coordinates with high absolute eigenvalues for each distribution set. 

(Page12, line418-line421) In this study, performance was evaluated by Performance Index (PI) defined by the sum of the squared error between the true probability mass function and the reconstructed probability mass function.

#############

Minor comments (7): Page 10, Line 344~345, please denote explicitly log q_i,j = log (inner product of P(x, theta^P), Q(x, theta^Q)).

Response: We added donations as below.

### Changes #######

(Page11, Method) Consider an n × n matrix M, whose (i, j)-th element m_ij is identified as , where qij is the functional inner product between i-th and j-th distributions.

##############

Minor comments (8): Supplement Page 3, Figure A. The order of figures and the legend description is not consistent. It should be “2D, Random, 1D, Mixture”

Response: We corrected as below.

### Changes #########

(S1 Text, Page3, Legend of Fig A) Fig A. Original parameter structure (first column), distribution (second column), θ coordinate mapping (third column), and boxplots of performance (fourth column) for four types of distribution set (2D, Random, 1D, Mixture). 

###################

Minor comments (9): Supplement Page 4, Line 5 in main text. Correct “… a larger a value of…” to “a larger value of …”

Response: We’re sorry it’s typo. We corrected as below.

### Changes #########

(S1 Text, Page 4) F_2(x) indicates that a larger value of ~

#####################

Minor comments (10): Supplement Page 5, Figure C. What is the x-axis in the plots and why is its support [0,1]? Please also label the y-axis.

Response: The range of discretization is scaled from 0 to 1. Part of the plot between 0.1 and 0.9 of the entire region were drawn as Mixture(sub). We revised the insufficient Figure legend.

### Changes #########

(S1 Text, Page4, Legend of Fig B): The range of discretization is scaled from 0 to 1. 

###################

Comments from Reviewer 2 

Thank you for the useful suggestions. We have highlighted the changes within the manuscript. The changed parts in the manuscript are also shown under the each response.

Comment(1): How does this work complement or expand on other algorithms offering similar approaches but more complex (RNA) data sets, for example Barkas, Nature Methods Aug. 2019? 

Response: 

Thank you for useful suggestion. 

Barkas et al.’s method, Conos integrates multiple single cell expression profiles into unified graph representation and identifies subpopulations, performs differential expression and annotates them. While both Conos and our DEEF take multiple single cell expression profiles as input, their purposes and outputs are completely different. First, Conos aims to cluster cell populations from multiple profiles by constructing a graph structure of the cells of all samples. The purpose of DEEF is completely different. In DEEF, each cell in a profile is considered as a sample from the probability distribution, the profile that is a distribution of cells, is represented as one point on the θ coordinate information geometric space. The all of the major functions of DEEF, such as drawing dynamics, analyzing data-driven differences, and time-course reconstruction, as shown in our paper, are outside the scope of Conos.

Although their purpose and outputs are quite different each other, they share an important concept; both are the methods to process multiple distributional profiles to data-mine. Therefore I quoted this article in the Introduction section.

### Changes #########

(Page2, Introduction, line23-line26) In recent years, demand for a computational method for heterogeneous multiple samples in the form of distribution has been increasing, and actually a method that integrates multiple expression profiles together and to identify subpopulation in data-driven manner was proposed [10].

######################

Comment(2): The authors discuss that their current workflow only works on relatively low dimensional data. Wetlab methods move away from low dimension towards more complex data, such as scRNA-seq or high dimensional maps of tissues or mass cytometry of complex blood and tissue samples. In order for this work to be applicable to a broader public the authors must use more complex data and also combine their analysis with dimension reduction methods such as UMAP (Uniform Manifold Approximation and Projection). UMAP runs very fast and could address many performance issues mentioned. Also UMAP, controversially to SPADE or the herein used trees, keeps single cells while giving the data a direction in form of population development or development over time.

Response: Thank you very much for very promising suggestion. As you mentioned our method was designed handle relatively low dimensional data and its novelty is to decompose a set of data samples in the form of distribution in data-driven fashion. The extension of our method to high-dimensional expression profiles by combining with dimension reduction method such as UMAP could bring a breakthrough for the limitation of our method. However, from information geometry standpoint, we feel careful evaluations would be necessary before applying our DEEF to distribution profiles in the non-linearly embedded coordinate systems. 

Some preliminary considerations will be as follow. First, when DEEF is performed after dimension reduction, the compositional distribution F is expressed as a distribution in the UMAP space. It is not easy to associate UMAP coordinate with the original gene expression. Since each cell has information on both UMAP coordinate values and the original gene expression values, that information may be used for the interpretation of F(x), but finding a good method might not be straightforward. Second, all cells in the dataset need to be applied UMAP simultaneously. This will increase the amount of computation as the number of samples increases even though DEEF itself is suitable for datasets with a large number of samples. Possibly, other or future dimension reduction method is more suitable. 

After all these issues were carefully considered, we revised as follow.

First, the title and abstract were changed to avoid misleading the readers to anticipate our method is easily applicable to high-dimensional profiles to “Decomposition of a set of distributions in extended exponential family form for distinguishing multiple oligo-dimensional marker expression profiles of single-cell populations and visualizing their dynamics” . And we revised the Abstract, Introduction to clarify this point. Also we added comments on the importance of development of the method that is applicable to high-dimensional cases instead of applying our method to those data sets in Discussion to clarify this point.

######### Changes #############

(Page1, Abstract)In this study, we propose a nonparametric statistical method, called Decomposition into Extended Exponential Family (DEEF), that embeds a set of single-cell expression profiles of several markers into a low-dimensional space and identifies the principal distributions that describe their heterogeneity.

(Page1, Introduction, line8-line11) Such single-cell expression data can be used to quantify or identify specific cell subsets based on the biomarkers. For example, specific lymphocyte subset (e.g. T cell and B cell subset) have been defined by the expression patterns of several cell surface protein markers [4,5].

(Page9, Discussion, line321-line340) In recent years, high-dimensional single cell expression data such as scRNA-seq or CyTOF has become popular. Computational methods for such high-dimensional single cell expression data are also being actively developed [27]. On the other hand, DEEF is not suitable for handling genome-wide gene expression because the number of grids grows exponentially with the dimensionality and kNN estimation and the linear algebraic algorithm can't work well. However, by the novel theory and algorithm, DEEF provides high-resolution analysis for sample heterogeneity where the calculated coordinates and the original marker expression pattern are completely associated by F(x) function.In many case, cellular subsets, such as lymphocyte subset, have been defined by the expression patterns of several markers. From this perspective, DEEF are expected to provide a novel insight on the analysis of cell population profiles. Then, it is necessary to select only a few important markers for high-dimensional CyTOF and scRNA-seq data. Although choosing irrelevant markers would theoretically not have much effect on the results because DEEF treats each grid as independent, it would waste computational resources. One potential solution might be the combination of DEEF with dimension reduction method, such as t-SNE and Uniform Manifold Approximation and Projection (UMAP) [28], although it seems necessary to study the effect of the non-linear embedding on the DEEF’s decomposition logic. Further investigations would be beneficial to overcome this drawback.

Several other improvements can be considered for the DEEF method. In its present form, DEEF handles grids independently; it does not consider the positional relationships among neighboring grid cells.

##################

---

## [Decision Letter · Decision Letter 1]

20 Mar 2020

Decomposition of a set of distributions in extended exponential family form for distinguishing multiple oligo-dimensional marker expression profiles of single-cell populations and visualizing their dynamics

PONE-D-20-01313R1

Dear Dr. Yamada,

We are pleased to inform you that your manuscript has been judged scientifically suitable for publication and will be formally accepted for publication once it complies with all outstanding technical requirements.

With kind regards,

Alan D Hutson

Academic Editor

PLOS ONE

Additional Editor Comments (optional):

Reviewers' comments:

Reviewer's Responses to Questions

**Comments to the Author**

1. If the authors have adequately addressed your comments raised in a previous round of review and you feel that this manuscript is now acceptable for publication, you may indicate that here to bypass the “Comments to the Author” section, enter your conflict of interest statement in the “Confidential to Editor” section, and submit your "Accept" recommendation.

Reviewer #1: (No Response)

Reviewer #2: All comments have been addressed

2. Is the manuscript technically sound, and do the data support the conclusions?

Reviewer #1: Yes

Reviewer #2: Yes

3. Has the statistical analysis been performed appropriately and rigorously? 

Reviewer #1: Yes

Reviewer #2: I Don't Know

4. Have the authors made all data underlying the findings in their manuscript fully available?

Reviewer #1: Yes

Reviewer #2: Yes

5. Is the manuscript presented in an intelligible fashion and written in standard English?

Reviewer #1: Yes

Reviewer #2: Yes

6. Review Comments to the Author

Reviewer #1: Most of my concerns have been addressed by the authors’ careful revision. However, the authors clarified that the proposed method can only be applied to relatively low dimensional single cell expression profiles. Therefore, my major concern still remains in that the method’s application in its current form is quite limited, even for data sets with moderate number of variables. In addition, the authors commented that "it is necessary to select only a few important markers for high-dimensional CyTOF and scRNA-seq data", but this is not always possible for complex data sets. Although the proposed method is interesting and may be significantly improved by combination with other dimension reduction methods as the other Reviewer suggested, it still requires further development. Therefore, based on the current form of this work, I would recommend rejection.

Reviewer #2: The authors have explained their incentives, pointed out the limitations of their approach, and addressed all my questions. Thank you.

7. PLOS authors have the option to publish the peer review history of their article (what does this mean?). If published, this will include your full peer review and any attached files.

Reviewer #1: No

Reviewer #2: No

---

## [Editor Report · Acceptance letter]

24 Mar 2020

PONE-D-20-01313R1 

Decomposition of a set of distributions in extended exponential family form for distinguishing multiple oligo-dimensional marker expression profiles of single-cell populations and visualizing their dynamics 

Dear Dr. Yamada:

I am pleased to inform you that your manuscript has been deemed suitable for publication in PLOS ONE. Congratulations! Your manuscript is now with our production department. 

With kind regards,

on behalf of

Dr. Alan D Hutson 

Academic Editor

PLOS ONE